# Racial/Ethnic Discrimination and Intimate Partner Violence Perpetration in Latino Men: The Mediating Effects of Mental Health

**DOI:** 10.3390/ijerph17218148

**Published:** 2020-11-04

**Authors:** Ana Isabel Maldonado, Carol B. Cunradi, Anna María Nápoles

**Affiliations:** 1Division of Intramural Research, National Institute on Minority Health and Health Disparities, National Institutes of Health, 9000 Rockville Pike, Building 3, Floor 5, Bethesda, MD 20892, USA; anna.napoles@nih.gov; 2Department of Psychology, University of Maryland Baltimore County, 1000 Hilltop Circle, Baltimore, MD 21250, USA; 3Prevention Research Center, Pacific Institute for Research and Evaluation, 2150 Shattuck Avenue, Suite 601, Berkeley, CA 94704, USA; cunradi@prev.org

**Keywords:** Latino, immigrant status, racial/ethnic discrimination, anxiety, depression, posttraumatic stress symptoms, alcohol dependence, drug dependence, intimate partner violence perpetration

## Abstract

Purpose: Intimate partner violence (IPV) is a serious public health problem that disproportionately affects racial/ethnic minorities in the U.S. This study examines risk factors for IPV perpetration that are salient for racial/ethnic minorities; specifically, we test if racial/ethnic discrimination among Latino men is associated with IPV perpetration, if poor mental health (MH) mediates this link, and whether relationships differ by immigrant status. Methods: Using National Epidemiologic Survey on Alcohol and Related Conditions (NESARC-II) Wave 2 (2004–2005) data, multigroup structural equation modeling compared immigrant (*N* = 1187) and U.S.-born (*N* = 1077) Latinos on a mediation model whereby discrimination increases IPV risk via poor MH (anxiety, depression, post-traumatic stress (PTSS); alcohol dependence (AD) and drug dependence (DD)). Results: For U.S.-born Latinos, discrimination increased anxiety (β = 0.24, *p* < 0.001), depression (β = 0.16, *p* < 0.001), PTSS (β = 0.09, *p* < 0.001), AD (β = 0.11, *p* < 0.001) and DD (β = 0.16, *p* < 0.001); anxiety (β = 0.16, *p* < 0.001), AD (β = 0.19, *p* < 0.001) and DD (β = 0.09, *p* < 0.01) increased IPV risk. Among Latino immigrants, discrimination increased anxiety (β = 0.07, *p* < 0.001), depression (β = 0.16, *p* < 0.001), PTSS (β = 0.08, *p* < 0.001) and DD (β = 0.03, *p* < 0.001); PTSS (β = 0.16, *p* < 0.001), AD (β = 0.21, *p* < 0.001) and DD (β = 0.05, *p* < 0.01) increased IPV risk. Conclusions: Among Latino men, discrimination is associated with poorer MH and contributes to IPV perpetration; MH risk factors vary by immigrant status.

## 1. Introduction

In the U.S., Latinos are an ethnic minority group that consist of Latin American immigrants and their descendants. As with other ethnic minorities, Latinos have faced multiple injustices throughout U.S. history ranging from segregation to lynchings to mass deportations [1,2,3], and continue to face discrimination today [4]. Latinos, specifically, Mexicans and their descendants, have been part of the U.S. since the mid-1800s following the Treaty of Guadalupe Hidalgo. However, it was not until recently that Latinos became a significant portion of the U.S. population. In 1966, Latinos made up only 4% of the total U.S. population [5]. That number has more than quadrupled, with Latinos making up nearly 18% of the total population in 2019 [6]. Although Latinos increased from 16% to 18% of the total population from 2010 to 2019, Latinos accounted for more than half (52%) of the population growth that occurred during this same time period [6] and are expected to continue accounting for a significant portion of population growth over the next few decades. In fact, Latinos are the largest ethnic minority group in the U.S., and they are expected to make up nearly 25% of the total population by 2065 [7]. Based on the current and projected population size and growth of U.S. Latinos, it is important to conduct research on public health problems that are relevant to this segment of the U.S. population; these include intimate partner violence (IPV).

In the U.S., IPV is a serious public health problem with 36% of women and 29% of men being raped, physically assaulted and/or stalked by their intimate partner at some point in their life [8]. The impact of IPV in the U.S. is not uniform and substantial racial/ethnic disparities in its effects have been reported [9]. For example, some studies find that Latinos are at elevated risk for IPV compared to whites [10,11], yet results of other studies are inconsistent [12,13].

Despite these inconsistencies, some research suggests that Latino couples are more likely to engage in male-perpetrated IPV than white couples [14,15]. Although female-perpetrated and bidirectional IPV are very prevalent [16,17,18], in this paper we focus on male-perpetrated IPV (henceforth referred to simply as IPV) due to its potential to be more injurious than female-perpetrated IPV [19]. Due to the increased risk of serious injury, it is critical to understand risk factors that contribute to IPV perpetration among Latino men.

Social ecological frameworks of IPV state that social structures (e.g., racism) can negatively impact an individual and increase their risk for IPV [20,21]. Therefore, racial/ethnic discrimination, an interpersonal form of racism, and poorer mental health (MH), a negative consequence of racism, may be salient to the study of IPV among U.S. Latinos. This line of research is particularly relevant to healthcare professionals, including mental health and primary care providers, who seek to screen and identify individuals at risk for IPV.

## 2. Racial/Ethnic Discrimination and Intimate Partner Violence Perpetration

Racial/ethnic discrimination has been identified as a risk factor for IPV. For instance, one study found that among young Latino adults, racial/ethnic discrimination was a risk factor for self-reported bidirectional IPV perpetration, and that the effect of discrimination on bidirectional IPV perpetration was stronger for immigrant than for U.S.-born Latinos [22]. Similarly, another study found that Black men who experienced higher levels of discrimination were more likely to perpetrate IPV than those who reported lower levels [23]; however, they did not distinguish between foreign-born and U.S.-born Black men. Therefore, it is not clear if the effect of racial/ethnic discrimination on IPV perpetration is the same for U.S.-born and foreign-born individuals. Research that examines the link between discrimination and IPV perpetration in Latino populations is limited. Moreover, research among Latinos has yet to elucidate whether the effect of racial/ethnic discrimination on IPV perpetration depends on immigrant status.

## 3. Mental Health and Intimate Partner Violence Perpetration

The link between MH and IPV perpetration is well established. A meta-analysis found that men’s physical assault perpetration was associated with depression, anxiety, and post-traumatic stress disorder (PTSD) [24]. This review did not examine the association of perpetration with alcohol or drug dependence. However, another meta-analysis has found a robust relationship between IPV perpetration and substance use, including alcohol dependence and drug dependence [25]. Moreover, a study based on data from the U.S. National Epidemiologic Survey of Alcohol and Related Conditions (NESARC) found that IPV perpetrators were more likely to report any MH disorder than non-perpetrators, including generalized anxiety, depression, PTSD, alcohol dependence and drug dependence [9]. No studies could be found that examined associations between MH and IPV among Latinos, specifically. However, one NESARC study did compare immigrants versus non-immigrants of varying races/ethnicities on the relationship of MH to IPV perpetration [26]. That study found that immigrants were more likely to perpetrate IPV than U.S.-born individuals and that immigrant perpetrators were more likely than immigrant non-perpetrators to have an MH disorder, including depression, anxiety, PTSD, an alcohol use disorder (abuse or dependence) and a drug use disorder (abuse or dependence). Among immigrants, Latinos had the highest rates of IPV perpetration. However, this study was limited in that it did not distinguish immigrants and non-immigrants by racial/ethnic background.

## 4. Mental Health as a Mediator

Though there is some research to suggest that racial/ethnic discrimination is a risk factor for IPV perpetration among minorities, there is no research to date that examines how these are linked. MH may serve as a mediator since it is a known key risk factor for IPV perpetration and a known outcome of racial/ethnic discrimination.

Previous research has established a relationship between racial/ethnic discrimination and MH in ethnic minorities. For instance, meta-analyses suggest that racial/ethnic discrimination is associated with depressive symptoms, anxiety symptoms, post-traumatic stress symptoms (PTSS) and substance use problems, including alcohol and drug abuse and dependence [27,28,29]. Furthermore, the effects of racial/ethnic discrimination on MH are persistent, being predictive of these MH outcomes more than one-year later [28]. These relationships did not vary by racial/ethnic group or birthplace.

Racial/ethnic discrimination has also been identified as a risk factor for poor MH among Latino populations in the U.S. For example, one study using a nationally representative sample of Latinos (National Latino and Asian American Study; NLAAS) found that Latinos who reported experiencing discrimination (compared to those who reported none) were more likely to suffer from anxiety disorders, including generalized anxiety and PTSD; mood disorders, including depression; and substance use disorders, including alcohol dependence and drug dependence [30]. Similarly, a meta-analysis focused on Latinos found a robust, positive association between racial/ethnic discrimination and unhealthy behaviors, such as substance use, as well as with depression and anxiety [31]. A daily diary study on Latinos provided evidence that racial/ethnic discrimination is a determinant of poor MH [32]. This study only examined depression symptoms. Immigrant status was not explored in any of these studies. Although immigrant status of Latinos is rarely considered in studies of discrimination and MH or IPV research, one study that did disaggregate U.S.-born and immigrant Latinos of Mexican descent found a stronger relationship between discrimination and depressive symptoms among immigrants than U.S.-born Latinos [33]. This suggests that it is important to disaggregate these two groups when examining MH as a potential mechanism (mediator) linking discrimination and IPV.

## 5. Rationale for Current Study

Social ecological frameworks of IPV indicate that social structures can negatively affect an individual and increase their risk for IPV perpetration. We posit that racism, a type of social structure, is a significant stressor that is contributing to IPV risk among Latinos. Racism manifests in a variety of ways. Here, we focus on an interpersonal form of racism, racial/ethnic discrimination. We suggest that discrimination, like many other stressors, will also increase risk for poor MH. In turn, we expect that poor MH will increase risk for IPV perpetration. We base this not only on prior research, but also on evidence that poor MH is characterized by emotion dysregulation (e.g., difficulty regulating one’s anger) and cognitive distortions (e.g., hostile attributions of one’s romantic partners actions), which can increase the risk of engaging in violence.

The current study aims to address several research gaps on the links between racial/ethnic discrimination, MH and IPV perpetration among Latino men. In addition to there being few of these studies, prior research has not examined how these relationships may vary by immigrant status. It is unknown how racial/ethnic discrimination affects risk of IPV perpetration among Latino immigrants, and whether the relationships between discrimination, MH and IPV vary by immigrant status. Therefore, this study, using secondary data analysis, aims to identify the relationships between discrimination, MH and IPV perpetration among Latino men, and whether these relationships differ by immigrant status. In the current study, five separate indicators of MH are included (generalized anxiety symptoms, depression symptoms, PTSS, alcohol dependence symptoms and drug dependence symptoms). These five indicators were included as they are some of the most consistently studied in the literature and they also have robust associations with both racial/ethnic discrimination and IPV perpetration.

Based on prior research in tandem with social ecological frameworks of IPV, we hypothesize:

**Hypothesis** **1 (H1).**
*Racial/ethnic discrimination will be associated with IPV perpetration for both U.S.-born and immigrant Latinos.*


**Hypothesis** **2 (H2).**
*Racial/ethnic discrimination will be associated with poorer MH, which in turn will be associated with IPV perpetration. These relationships are expected to be stronger for Latino immigrants than for U.S.-born Latinos.*


**Hypothesis** **3 (H3).**
*MH will mediate the association between racial/ethnic discrimination and IPV perpetration.*


Results from this study would help clinicians in primary care and mental health practitioners develop appropriate screening tools to identify groups at a particularly high risk of IPV perpetration. Knowing whether discrimination has direct effects and/or indirect effects on risk of IPV perpetration, which specific mental health symptoms may mediate the effects of discrimination on IPV perpetration, and how risk of IPV perpetration varies by immigrant status could begin to characterize which specific population groups may need more targeted screening and interventions to prevent IPV.

## 6. Methods

### 6.1. Participants

This secondary data analysis study used cross-sectional data from Wave 2 (2004–2005) of the National Epidemiologic Survey on Alcohol and Related Conditions (NESARC). The NESARC employed a multistage cluster sampling design to obtain a nationally representative sample of noninstitutionalized U.S. adults, 18 years or older. Blacks, Latinos and young adults were oversampled in order to obtain reliable estimates for these groups. More details on the study design sampling and data collection can be found elsewhere [34,35]. The current study restricted the sample to Latino men who reported being in an intimate relationship in the past year (*N* = 2287; *n* = 1092 for Latino U.S.-born men; and *n* = 1195 for Latino immigrant men). Table 1 reports the estimated population parameters based on the sampling design of the survey. Here, we report raw sample data without weighting. In the current sample, participants were, on average, approximately 43 years old. Almost half had at least some college education (46.2%), 24.3% had a high school education or equivalent, and 29.6% had less than a high school education. Most identified as heterosexual or straight (97.60%). Slightly less than half were born in the U.S. (48%). Immigrants had lived, on average, 22.40 years (*SD* = 12.92) in the U.S. Overall, 12.9% of Latino male participants endorsed at least one depressive symptom, 4.9% endorsed at least one anxiety symptom, 6.7% endorsed at least one PTSS, 31.8% endorsed at least one alcohol dependence symptom, and 5.0% endorsed at least one drug dependence symptom. In this sample, the following percentage of participants met criteria for a probable mental disorder diagnosis: 6.2% for major depressive disorder, 2.3% for generalized anxiety disorder, 5.1% for PTSD, 9.5% for alcohol dependence disorder, and 3.6% for drug dependence disorder. IPV perpetration was endorsed by 5% of immigrants and 6% of U.S. born Latino men.

### 6.2. Measures

#### 6.2.1. Racial/Ethnic Discrimination

Racial/ethnic discrimination was measured using a modified 6-item Experiences of Discrimination scale [36]. Participants were asked to report how frequently, from “never” (0) to “very often” (4), they experienced discrimination in the following situations: (1) obtaining health insurance; (2) receiving care; (3) in public settings (on the street, in stores or restaurants); (4) obtaining a job, on the job, getting admitted to school/training program, in the courts or by police, or obtaining housing; (5) being called a racist name; and (6) being made fun of, picked on, pushed, shoved, hit or threatened with harm. Participants reported on discrimination for two time periods: during the past year and during the 1–2-year time interval between the prior interview (wave 1) and the past year. The racial/ethnic discrimination score was created by averaging responses to these items across both time periods. Eight participants were coded as missing because they did not have a valid response for at least 75% of the scale items. The scale had excellent reliability in both U.S.-born and immigrant Latino men (Cronbach’s α = 0.91 and 0.89, respectively); scale scores ranged from 0 to 4, with higher scores indicating more discrimination.

#### 6.2.2. Mental Health

Five measures of MH were assessed: generalized anxiety symptoms, depression symptoms, post-traumatic stress symptoms, alcohol dependence symptoms and drug dependence symptoms. Each measure had symptom questions corresponding to the Diagnostic and Statistical Manual of Mental Disorders (4th ed.; DSM-IV) diagnostic criteria [37]. The NESARC mental health assessment methods and reliability are described in more detail elsewhere [34,38].

A summary score for each of the five MH measures was calculated as the sum of endorsed symptoms since the last interview (wave 1), which corresponds to the past 2–3 years. This timeframe is consistent with the timeframe for the discrimination measure. Scores were only calculated when a participant answered the screening question affirmatively, if applicable, and provided a valid response for at least 75% of the symptom items. As a result, responses were coded as missing for 3 participants for PTSS and 1 for drug dependence; there were no responses coded as missing for depression, generalized anxiety or alcohol dependence.

##### Anxiety Symptoms

To assess generalized anxiety, participants who endorsed feeling “tense, nervous, or worried most of the time” for at least 6 months were asked whether they had experienced 15 anxiety symptoms since the prior interview. The scale had excellent reliability in both U.S.-born and immigrant Latino men (Cronbach’s α = 0.97 and 0.96, respectively); scale scores ranged from 0 to 15, with higher scores indicating more anxiety symptoms.

##### Depression Symptoms

To assess depression, participants who endorsed feeling “sad, blue, depressed or down most of the time” or “didn’t care about things [they] usually cared about” for at least two weeks were asked whether or not they had experienced 19 depression symptoms since the prior interview. The scale had excellent reliability in both U.S.-born and immigrant Latino men (Cronbach’s α = 0.94 for both); scale scores ranged from 0 to 19, with higher scores indicating more depressive symptoms.

##### Posttraumatic Stress Symptoms

To assess posttraumatic stress, participants who endorsed a Criterion A traumatic event (exposed to/witnessed a death, threatened death, actual or threatened serious injury, or actual or threatened sexual violence) were asked whether they had experienced 19 PTSS since the prior interview (past 2–3 years). The scale had excellent reliability in both U.S.-born and immigrant Latino men (Cronbach’s α = 0.92 for both); scale scores ranged from 0 to 19, with higher scores indicating more PTSS.

##### Alcohol Dependence Symptoms

To assess alcohol dependence, participants were asked whether they had experienced 17 alcohol dependence symptoms since the prior interview. The scale had good reliability in both U.S.-born and immigrant Latino men (Cronbach’s α = 0.86 and 0.85, respectively); scale scores ranged from 0 to 17, with higher scores indicating more alcohol dependence symptoms.

##### Drug Dependence Symptoms

To assess drug dependence, participants were asked whether they had experienced 13 drug dependence symptoms since the prior interview. The scale had excellent reliability in both U.S.-born and immigrant Latino men (Cronbach’s α = 0.92 and 0.94, respectively); scale scores ranged from 0 to 13, with higher scores indicating more drug dependence symptoms.

#### 6.2.3. Intimate Partner Violence Perpetration

IPV perpetration in the past year was assessed via 6 items that are a combination of multiple items from the Conflict Tactics Scale (CTS), Form R with modified response options [39]. This approach is consistent with other research on IPV that has been published using the NESARC [9,18,26]. Participants who endorsed being in a relationship were asked how frequently in the past year (0 = never to 5 = more than once a month) they: (1) pushed, grabbed, or shoved; (2) slapped, kicked, bit, or hit; (3) threatened with a weapon; (4) cut or bruised; (5) forced sex; and (6) injured their spouse/partner. Participants were categorized as perpetrators (1) or non-perpetrators (0). Twelve participants did not have valid responses to any of the CTS items, and were coded as missing.

#### 6.2.4. Immigrant Status

Immigrant status was assessed by a self-reported item asking whether participants were born in the U.S. or not.

#### 6.2.5. Covariates

Two covariates, socio-economic status and acculturation, were adjusted for in the present study given their links with MH [40,41] and IPV perpetration [9,42,43]. Education level (continuous) was used as a proxy for socio-economic status. To assess for U.S. acculturation, the 7-item Language Orientation subscale of the Short Acculturation Scale was used [44]. Language-based acculturation was used in the present study instead of other dimensions of acculturation because this dimension is robustly associated with negative MH [41]. This measure had excellent reliability in both U.S.-born and immigrant Latino men (Cronbach’s α = 0.90 and 0.91, respectively). Lower scores for language-based acculturation reflect predominantly speaking Spanish whereas higher scores reflect predominantly speaking English.

#### 6.2.6. Data Analysis

The lavaan structural equation modeling package [45] in R 3.6.1 [46] was used to: run descriptive statistics and bivariate correlations; and test a multigroup structural equation model (SEM) with parallel mediators wherein racial/ethnic discrimination predicts each of the five MH measures, and these five in turn predict IPV; and examine whether these paths varied by immigrant status (Figure 1). Multigroup SEM is recommended over traditional mediation analysis because of its ability to test for group differences in the significance of indirect effects in a single analysis [47]. Another advantage of the SEM approach is that it provides several model fit indices to assess the fit of hypothesized mediation models. To test for path differences across immigrant status, a χ^2^ test of relative fit, *p* < 0.05, was used to compare two models; one where all paths were constrained to be equal across immigrant status and one where the paths were allowed to vary freely. To determine whether the final model exhibited reasonable fit, the following recommended cut-off values were used: Comparative Fit Index (CFI) and Tucker Lewis Index (TLI) greater than 0.95, Standardized Root Mean Square Residual (SRMR) less than 0.08, and Root Mean Square Error of Approximation (RMSEA) less than 0.05 [48,49]. The complex survey design was taken into account for all analyses (i.e., descriptive statistics, correlations and multigroup parallel mediation modeling) using the lavaan survey package [50]. Given the complex survey design, the indirect effects were tested using Monte Carlo 95% confidence intervals [51]. Since the outcome was dichotomized, a probit link function was utilized. Non-normality was handled using diagonally weighted least squares (DWLS), a robust version of weighted least squares [52]. In the final model, 15 U.S.-born and 8 immigrant Latinos had missing data. These observations were dropped through listwise deletion given that this is the default (and only) option in lavaan when using the DWLS estimator.

## 7. Results

### 7.1. Correlations

Table 2 presents the correlation matrix for U.S.-born Latinos (upper half) and Latino immigrants (lower half). Among U.S.-born Latinos, racial/ethnic discrimination had positive, significant correlations with all MH measures (i.e., anxiety symptoms, depression symptoms, PTSS, alcohol dependence symptoms and drug dependence symptoms) and IPV perpetration. Racial/discrimination had a negative, significant correlation with language-based acculturation, such that higher discrimination was associated with predominantly speaking Spanish. Racial/ethnic discrimination had no significant correlation with education. The MH measures had positive, significant correlations with each other, except there were no significant correlations between anxiety and alcohol dependence nor anxiety and drug dependence. IPV perpetration had a positive, significant correlation with all MH measures except PTSS. IPV perpetration had no significant correlation with education and had a positive, significant correlation with language-based acculturation, such that IPV perpetration was associated with predominantly speaking English in U.S.-born Latinos.

Among Latino immigrants, racial/ethnic discrimination had positive, significant correlations with PTSS, depression symptoms and language-based acculturation, such that higher discrimination was associated with predominantly speaking English. Racial/ethnic discrimination had no significant correlations with the other MH measures, IPV perpetration or education. The MH measures had positive, significant correlations with each other. However, there were no significant correlations between anxiety and the substance use measures (i.e., alcohol dependence and drug dependence), and between PTSS and the substance use measures. IPV perpetration had a significant, positive correlation with PTSS, alcohol dependence symptoms, and drug dependence symptoms. There was a significant, negative correlation between IPV perpetration and education. Finally, there was a significant, positive correlation between IPV perpetration and language-based acculturation, such that Latino immigrants who predominantly speak English were more likely to engage in IPV perpetration.

### 7.2. Multi-Group Structural Equation Model

Figure 1 and Figure 2 display the unstandardized and standardized multigroup parallel mediational model results; all paths are adjusted for education and language-based acculturation. The hypothesized model appears to be an excellent fit to the data (CFI = 1.00, TLI = 1.00, RMSEA = 0.00, SRMR = 0.00). This is a significant improvement over the fully constrained model where all paths were set to be equal across groups (CFI = 0.96, TLI = 0.88, RMSEA = 0.09, SRMR = 0.04). This significant improvement is demonstrated by the test of path differences, which revealed that at least one of the paths varied by immigrant status, ∆χ^2^(23, *N* = 2264) = 220.30, *p* < 0.001. The results are presented below and organized by MH measure.

### 7.3. Anxiety Symptoms

First, for both U.S.-born and immigrant Latino men, higher levels of racial/ethnic discrimination were associated with more anxiety symptoms; however, this relationship was stronger, ∆χ^2^(1, *N* = 2264) = 38.44, *p* < 0.001, for U.S.-born Latinos, β = 0.24, *z* = 8.61, *p* < 0.001, than for Latino immigrants, β = 0.07, *z* = 4.25, *p* < 0.001. Second, the association between anxiety symptoms and IPV perpetration varied by immigrant status, ∆χ^2^(1, *N* = 2264) = 9.55, *p* = 0.002. Higher levels of anxiety symptoms were associated with a greater likelihood of IPV perpetration only among U.S.-born Latinos, β = 0.16, *z* = 5.71, *p* = 0.016, and not Latino immigrants, β = 0.02, *z* = 1.21, *p* = 0.228. Finally, there was a significant indirect effect through anxiety symptoms for U.S.-born Latinos, *ab* = 0.022, 95% CI (0.013, 0.031), but not for Latino immigrants, *ab* = 0.0001, 95% CI (−0.0012, 0.0015); and the indirect effect for U.S.-born Latinos was stronger (see Table 3 for indirect effects).

### 7.4. Depression Symptoms

For both U.S.-born and immigrant Latino men, higher levels of racial/ethnic discrimination were associated with more depression symptoms; however, this relationship was stronger, ∆χ^2^(1, *N* = 2264) = 7.42, *p* = 0.006, for U.S.-born Latinos, β = 0.163, *z* = 8.17, *p* < 0.001, than for Latino immigrants, β = 0.155, *z* = 6.14, *p* < 0.001. Regardless of immigrant status, there was no significant relationship between depression symptoms and IPV perpetration, β = −0.03, *z* = −1.88, *p* = 0.060. As a result, there was no significant indirect effect through depressive symptoms for either U.S.-born Latinos, *ab* = −0.0030, 95% CI (−0.0065, 0.0001), or Latino immigrants, *ab* = −0.0031, 95% CI (−0.0068, 0.0002).

### 7.5. Posttraumatic Stress Symptoms

Regardless of immigrant status, Latino men experiencing higher levels of racial/ethnic discrimination had higher PTSS than those experiencing lower levels of discrimination, β = 0.09, *z* = 5.81, *p* < 0.001. Moreover, the association between PTSS and IPV perpetration varied by immigrant status, ∆χ^2^(1, *N* = 2264) = 7.07, *p* = 0.008. Higher levels of PTSS were associated with a greater likelihood of IPV perpetration, only among Latino immigrants, β = 0.16, *z* = 4.49, *p* < 0.001, and not U.S.-born Latinos, β = −0.01, *z* = −0.71, *p* = 0.476. Finally, there was a significant indirect effect through PTSS for Latino immigrants, *ab* = 0.009, 95% CI (0.005, 0.015), but not for U.S.-born Latinos, *ab* = −0.0007, 95% CI (−0.0028, 0.0012); and the indirect effect for Latino immigrants was stronger.

### 7.6. Alcohol Dependence Symptoms

The association between racial/ethnic discrimination and alcohol dependence varied by immigrant status, ∆χ^2^(1, *N* = 2264) = 6.62, *p* = 0.010. Higher levels of racial/ethnic discrimination were associated with more alcohol dependence symptoms only among U.S.-born Latinos, β = 0.11, *z* = 4.97, *p* < 0.001 and not Latino immigrants, β = 0.02, *z* = 0.86, *p* = 0.393. For both U.S.-born and immigrant Latino men, higher levels of alcohol dependence were associated with a greater likelihood of IPV perpetration; however, this relationship was stronger, ∆χ^2^(1, *N* = 2264) = 5.01, *p* = 0.025, for Latino immigrants, β = 0.21, *z* = 7.24, *p* < 0.001, than for U.S.-born Latinos, β = 0.19, *z* = 5.82, *p* < 0.001. Overall, there was a significant indirect effect through alcohol for U.S.-born Latinos, *ab* = 0.012, 95% CI (0.007, 0.019), but not for Latino immigrants, *ab* = 0.002, 95% CI (−0.003, 0.008); however, the intervals overlap suggesting that the indirect effects did not differ.

### 7.7. Drug Dependence Symptoms

First, for both U.S.-born and immigrant Latino men, higher levels of racial/ethnic discrimination were associated with more drug dependence symptoms; however, this relationship was stronger, ∆χ^2^(1, *N* = 2264) = 28.09, *p* < 0.001, for U.S.-born Latinos, β = 0.16, *z* = 6.41, *p* < 0.001, than for Latino immigrants, β = 0.03, *z* = 3.83, *p* < 0.001. Second, regardless of immigrant status, higher levels of drug dependence were associated with a greater likelihood of IPV perpetration, β = 0.05, *z* = 3.43, *p* = 0.001. Finally, there was a significant indirect effect through drug dependence for both U.S.-born Latinos, *ab* = 0.008, 95% CI (0.003, 0.014), and Latino immigrants, *ab* = 0.0012, 95% CI (0.0004, 0.0023), but the indirect effect for U.S.-born Latinos was stronger.

### 7.8. Direct Effect

Regardless of immigrant status, there was no significant direct effect of racial/ethnic discrimination on IPV perpetration after adjusting for the five MH measures and covariates, β = 0.02, *z* = 1.21, *p* = 0.228.

## 8. Discussion

The current study examined the relationships among racial/ethnic discrimination, five MH measures (i.e., anxiety symptoms, depressive symptoms, PTSS, alcohol dependence symptom and drug dependence symptoms) and IPV perpetration in Latino men and whether these relationships varied by immigrant status. In the present study, we found that overall, as expected, positive associations of racial/ethnic discrimination and risk for perpetrating IPV were mediated by poorer MH. However, the specific mechanisms linking discrimination to IPV perpetration varied as a function of immigrant status. In particular, PTSS and drug dependence symptoms were important mechanisms that linked discrimination to IPV perpetration in Latino immigrants; whereas in U.S.-born Latinos, these mechanisms were anxiety symptoms, alcohol dependence symptoms and drug dependence symptoms.

A significant correlation between discrimination and IPV perpetration was observed among U.S.-born Latinos, but not in Latino immigrants. This was inconsistent with the one study we found on the link between discrimination and IPV in Latinos [22]. This study showed that discrimination was associated with IPV perpetration in both groups; additionally, because our hypotheses were based on this study, these findings were contrary to our expectation. Notably, the significant association for U.S.-born Latinos disappeared after adjusting for MH symptoms as demonstrated by a non-significant direct effect of discrimination on IPV perpetration, which was the same in both groups.

Notably, in line with our expectations, racial/ethnic discrimination had a detrimental effect on MH in both U.S.-born and immigrant groups. This is consistent with prior research on the robust relationship between racial/ethnic discrimination and numerous MH outcomes in Latinos and other racial/ethnic minorities [27,28,29]. Surprisingly, discrimination was not associated with alcohol dependence symptoms in Latino immigrants. This was the only non-significant finding between discrimination and MH in both groups. It is possible that there are more salient social stressors or cultural factors other than racial/ethnic discrimination that increase the risk for alcohol dependence symptoms in Latino immigrants.

Although the negative impact of racial/ethnic discrimination on MH was present in both groups, this impact of discrimination was consistently more detrimental to the MH of U.S.-born Latinos than the MH of Latino immigrants. This finding was contrary to our hypotheses as well as previous research that found that this relationship was stronger for Latino immigrants than U.S.-born Latinos [33]. The current findings were also inconsistent with a meta-analysis which found that there were no differences between immigrants and native-born individuals [28]. However, this inconsistency in findings may be due to the fact that the meta-analysis was not specific to the Latino population in the U.S. It was out of the scope of the current study to examine why these differences exist between immigrants and U.S.-born Latinos in the discrimination–MH link. However, these findings are consistent with the immigrant paradox literature [53] that shows that immigrants tend to experience “positive” outcomes despite the presence of risk factors. As suggested by the immigrant paradox literature, it is possible that there are cultural factors, such as social support [31], that may be helping Latino immigrants cope better, or more effectively, when faced with discrimination than U.S.-born Latinos.

Our findings are consistent with our hypotheses and established research on MH conditions as a risk factor for IPV perpetration [9,24]. They are also, generally, consistent with research on this topic among immigrants. For instance, one study [26] found that all the MH measures, which included the five measures in the current study, were associated with an increased risk of IPV perpetration. However, our findings suggested that only certain MH symptoms (i.e., PTSS, alcohol dependence and drug dependence) increased risk for IPV perpetration among Latino immigrants. These differences in findings, despite utilizing the same NESARC dataset, may be because these authors did not stratify by race/ethnicity which likely obscured any unique findings among Latino immigrants, specifically.

Although MH generally was associated with IPV perpetration in both groups, the MH symptoms that increased risk for IPV perpetration in U.S.-born Latinos were different from those of Latino immigrants. Specifically, for U.S.-born Latinos, although depressive symptoms were associated with IPV perpetration, only anxiety, alcohol dependence and drug dependence symptoms were uniquely associated with an increased risk of IPV perpetration. Some of these MH symptom risk factors are shared with Latino immigrants (e.g., anxiety, alcohol dependence and drug dependence); even so, the strength of the relationships varied by immigrant status.

Finally, in our study, Latino men with more experiences of racial/ethnic discrimination were at a greater risk for perpetrating IPV due to poor MH, but the specific mediating MH risk factors and strength of associations varied by immigrant status. In particular, PTSS and drug dependence symptoms were important mechanisms that linked discrimination to IPV perpetration in Latino immigrants; whereas in U.S.-born Latinos, these mechanisms were anxiety symptoms, alcohol dependence symptoms and drug dependence symptoms. This suggests that IPV prevention and intervention efforts that aim to address the impacts of racism on IPV risk in Latino men should consider targeting mechanisms based on whether the population is U.S.-born or foreign-born. Doing so may result in more effective prevention and intervention. For example, in addition to alcohol dependence symptoms, clinicians and practitioners providing services to IPV-involved Latino immigrants should also assess and treat PTSD symptoms as these appeared to be one of the stronger predictors of IPV perpetration in this population. In doing so, these practitioners should be mindful to consider trauma in the context of migration, which can occur before, during and after migration [54].

On the other hand, when working with U.S.-born Latinos, in addition to assessing and treating alcohol dependence symptoms, practitioners should also consider anxiety as this appeared to be the next strongest predictor of IPV perpetration in this population. Importantly, alcohol dependence symptoms were the strongest MH risk factor for IPV perpetration in both groups. This is consistent with previous research that has identified substance use as a key risk factor for IPV perpetration among Latino men [55]. Therefore, it is important for practitioners to assess for substance use in the prevention and intervention of IPV perpetration among Latino men.

### 8.1. Limitations

The present study has several limitations. First, the current study is cross-sectional, which precludes causal inferences. Although the current model is based on associations observed in prior longitudinal research, further longitudinal studies need to examine the directionality of the current findings, especially in the context of IPV. Second, the current sample consists of only Latino men. Future research should examine Latina women’s use of aggression considering prior research has found similar IPV rates to that of Latino men [56]. Third, although Latinos share many cultural aspects, they are a heterogenous group, whose beliefs and attitudes can vary by country of origin. The current SEM model was unable to examine these differences due to lack of power, but future research should focus on country of origin since IPV rates appear to be more prevalent in Latino immigrants from Central America, the Caribbean and Mexico [26].

### 8.2. Implications

The present study has implications for prevention and intervention efforts that occur both in the community and clinical settings, including primary care settings, general mental health clinics as well as relationship violence intervention programs (RVIPs), commonly referred to as Batterer Intervention Programs (BIPs). Some of these implications, such as tailoring prevention and interventions efforts to the target population (e.g., U.S.-born versus immigrants), have already been discussed above. In primary care settings, practitioners can screen and identify Latino men at risk for IPV paying particular attention to the fact that different mental health conditions are risk factors for U.S.-born Latinos and immigrants. Practitioners at mental health clinics that are treating Latinos should be mindful that discrimination (among other race-related traumas and stressors) may be contributing to and exacerbating their Latino clients’ mental health [57,58]. Therefore, clinicians should assess for racial discrimination and other racial stressors during intake assessments and integrate these assessments into case formulations and treatment planning. When clients present with psychological distress as a result of racial stressors, clinicians can integrate one of many clinical models that have been developed to treat racial trauma and other race-related stressors into their treatment plans [59,60,61].

The aforementioned implications are relevant not only for providers at MH clinics that serve Latino communities, but also to providers of RVIPs with substantial Latino populations. Therefore, practitioners at RVIPs should also consider the role of discrimination, and other race-based stressors [62], as potential risk factors for IPV perpetration among Latino men receiving services at RVIPs. Specifically, RVIP providers should make concerted efforts to address racial stressors and traumas by integrating racial trauma treatment approaches when working with Latino men [63]. In fact, these racial trauma approaches dovetail nicely with current trauma-informed approaches in RVIPs [64].

Moreover, although the present findings conceptualized discrimination as an interpersonal form of racism, the impact of structural forms of racism on IPV perpetration may be particularly important for prevention and intervention efforts to address. Although this is a daunting task, such efforts can reduce structural disparities that contribute to IPV perpetration, such as unemployment, by providing access to community resources. For instance, the Gateway Project at the House of Ruth, a U.S. RVIP in Baltimore, Maryland, offers IPV-involved clients employment support as one of many services provided to reduce risk associated with structural disparities. More broadly, the Gateway Project attempts to address systemic racism by incorporating treatment sessions that cover the role of systemic racism in the use of IPV [65].

The current findings also have important research implications. First, given our disparate findings based on immigrant status, researchers are cautioned against grouping U.S.-born and immigrant populations in analyses without first confirming that they do not differ. Second, the current research raises an important question that should be addressed by future research: why are IPV perpetration risk factors and mechanisms in Latino immigrants different from U.S.-born Latinos? It is possible, as discussed earlier, that some factors are more salient in one group versus the other. For instance, PTSD symptoms among immigrants may be more related to IPV perpetration as immigrants may experience more severe forms of PTSD by virtue of being more likely to be exposed to more traumatic experiences, including fleeing one’s home country due to political violence.

Furthermore, racism is multi-level, interpersonal, structural, and multi-faceted [66], and can vary in frequency, intensity, and severity [67]. In the current study, only one dimension of racism was examined. Future research should examine different aspects of racism and other race-related traumas and stressors and their role in the risk of IPV perpetration among Latinos.

Another question that arises from the current findings is what protective factors, especially culturally relevant ones, may reduce the overall negative effect of racial/ethnic discrimination on Latinos’ IPV perpetration. A meta-analysis suggests that social support mitigates the effect of racial/ethnic discrimination on MH outcomes among Latinos [31], but immigrant status was not accounted for. Future research should examine whether social support and other culturally relevant protective factors can mitigate the effect of racial/ethnic discrimination on IPV perpetration in Latinos.

## 9. Conclusions

Based on our findings, we conclude that racial/ethnic discrimination has a significant and harmful effect on MH, among both U.S.-born and immigrant Latinos, but especially among U.S.-born Latinos. This highlights the need to address discrimination in the mental health treatment of Latino populations and to differentiate between U.S.-born and immigrant Latinos, as the impact of discrimination may vary across these groups. In addition, our results highlight the need to distinguish between various mental health conditions and symptomatology in assessing for risk of IPV perpetration as the specific MH mechanisms linking discrimination to IPV perpetration can vary depending on immigrant status. Specifically, PTSS and drug dependence symptoms were important mechanisms that linked discrimination to IPV perpetration in Latino immigrants; whereas in U.S.-born Latinos, these mechanisms were anxiety symptoms, alcohol dependence symptoms and drug dependence symptoms. These key findings can help inform tailored IPV prevention and intervention efforts among Latino men. In particular, these findings shed light on the fact that MH mechanisms differ by immigrant status; and thus, tailoring and targeting IPV risk factors based on immigrant status in clinical and community settings may result in more effective IPV prevention interventions among this higher risk population.

## Figures and Tables

**Figure 1 ijerph-17-08148-f001:**
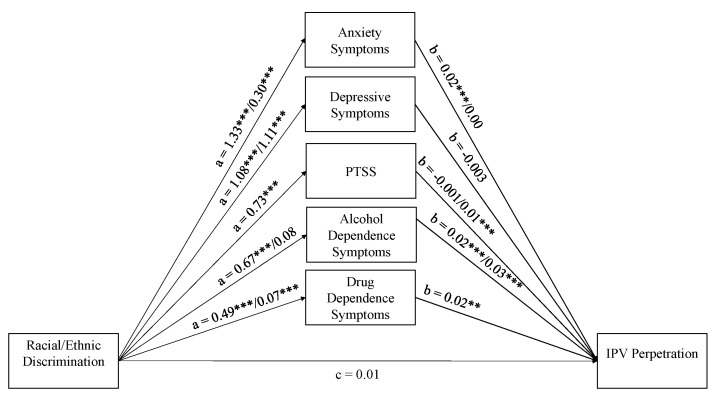
Unstandardized Multi-group Parallel Mediation Model. Note. PTSS = post-traumatic stress symptoms. IPV = intimate partner violence. All paths are unstandardized and adjusted for education-level and language-based acculturation. If the paths significantly varied across groups, then two estimates are provided. The first corresponds to U.S.-born Latinos and the second to Latino immigrants. Model fit: CFI = 1.00, TLI = 1.00, RMSEA = 0.00, SRMR = 0.00. *** p* < 0.01, *** *p* < 0.001.

**Figure 2 ijerph-17-08148-f002:**
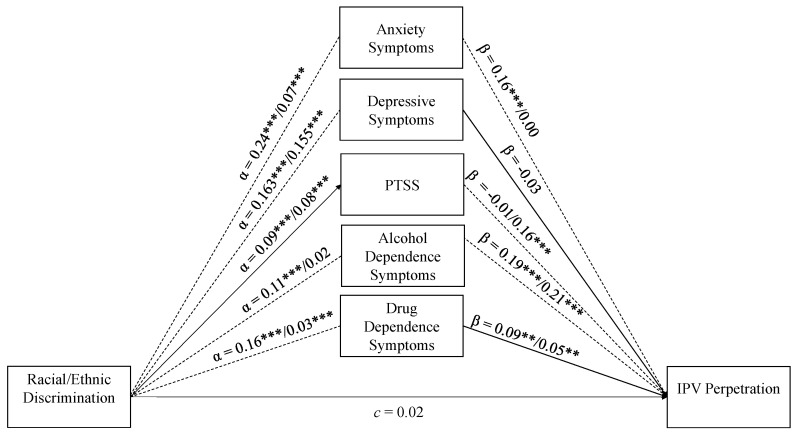
Standardized Multi-group Parallel Mediation Model. Note. PTSS = post-traumatic stress symptoms. IPV = intimate partner violence. All paths are standardized and adjusted for education-level and language-based acculturation. If the paths significantly varied across groups, then a dashed line is used. Regardless of whether the groups varied or not, two estimates are provided. The first corresponds to U.S.-born Latinos and the second to Latino immigrants. Model fit: CFI = 1.00, TLI = 1.00, RMSEA = 0.00, SRMR = 0.00. *** p* < 0.01, *** *p* < 0.001.

**Table 1 ijerph-17-08148-t001:** Demographics and Bivariate Differences by Immigrant Status at Population Level.

	Overall Sample (*N* = 2287)	Latino Immigrants (*n* = 1187)	U.S.-Born Latinos (*n* = 1077)	
	%	%	%	*p* ^b^
Less than high school ^a^	35.3%	48.5%	17.1%	<0.001
High school or equivalent ^a^	24.4%	22.3%	27.3%	*ns*
At least some college ^a^	40.2%	29.2%	55.6%	<0.001
	*M* (*SE*)	*M* (*SE*)	*M* (*SE*)	*p*
Age	40.54 (0.52)	41.00 (0.79)	40.04 (0.65)	*ns*
Language-based Acculturation	2.99 (0.06)	2.15 (0.04)	4.16 (0.05)	<0.001
Discrimination	0.18 (0.02)	0.16 (0.02)	0.22 (0.02)	0.024
Anxiety Symptoms	0.38 (0.06)	0.25 (0.05)	0.55 (0.12)	0.021
Depressive Symptoms	0.83 (0.08)	0.72 (0.09)	0.99 (0.14)	*ns*
PTSS	0.52 (0.07)	0.49 (0.11)	0.55 (0.08)	*ns*
Alcohol dependence symptoms	1.05 (0.07)	0.78 (0.08)	1.41 (0.10)	<0.001
Drug dependence symptoms	0.16 (0.03)	0.07 (0.02)	0.28 (0.05)	<0.001
IPV Perpetration ^a^	0.07 (0.01)	0.05 (0.22)	0.06 (0.23)	*ns*

Note: Values except sample size are weighted for study design; IPV = intimate partner violence; PTSS = post-traumatic stress symptoms. Language-based acculturation is coded such that lower scores reflect predominantly speaking Spanish whereas higher scores reflect predominantly speaking English. ^a^ Variable is categorical or dichotomous. ^b^
*t*-test for group differences between immigrants and U.S.-born Latinos.

**Table 2 ijerph-17-08148-t002:** Observed Correlation Matrix by Immigrant Status.

	RD	AS	DS	PTSS	ADS	DDS	IPV ^a^	Edu	LBA
Racial Discrimination (RD)		0.24 ***	0.16 *	0.12 ***	0.10 ***	0.15 ***	0.07 *	0.02	−0.09 **
Anxiety Symptoms (AS)	0.07		0.45 ***	0.34 ***	0.05	0.06	0.15 **	−0.02	0.01
Depression Symptoms (DS)	0.16 ***	0.37 ***		0.25 ***	0.14 **	0.17 *	0.09 **	−0.01	0.05
Post-traumatic Stress Symptoms (PTSS)	0.07 *	0.15 *	0.22 ***		0.09 *	0.15 **	0.06	0.02	−0.02
Alcohol Dependence Symptoms (ADS)	0.02	0.07	0.08 *	0.08		0.47 ***	0.23 ***	−0.04	0.07 *
Drug Dependence Symptoms (DDS)	0.05	0.05	0.34 ***	0.02	0.17 ***		0.19 ***	−0.09 ***	0.01
Intimate Partner Violence ^a^ (IPV)	0.05	0.04	0.03	0.18 ***	0.24 ***	0.08 *		−0.01	0.06 *
Education (Edu)	0.05	0.03	0.00	0.00	−0.01	0.03 *	−0.05 *		0.32 ***
Language-based Acculturation (LBA)	0.12 ***	0.04	0.05	0.02	0.06	0.10 **	0.06 *	0.48 ***	

Note. The lower half is the correlation matrix for Latino immigrants (*N* = 1187). The upper half is the correlation matrix for U.S.-born Latinos (*N* = 1077). ^a^ Variable is dichotomous. * *p* < 0.05, *** p* < 0.01, *** *p* < 0.001.

**Table 3 ijerph-17-08148-t003:** Indirect Effects of Mental Health Symptoms by Immigrant Status.

	U.S.-Born Latinos			Latino Immigrants		
Variable	Indirect Effect	Lower CI ^a^	Upper CI ^a^	Indirect Effect	Lower CI ^a^	Upper CI ^a^
Anxiety	0.022	0.013	0.031	0.0001	−0.0012	0.0015
Depression	−0.0030	−0.0065	0.0001	−0.0031	−0.0068	0.0002
PTSS	0.009	0.005	0.015	−0.0007	−0.0028	0.0012
Alcohol Dependence	0.012	0.007	0.019	0.002	−0.003	0.008
Drug Dependence	0.008	0.003	0.014	0.0012	0.0004	0.0023

Note: PTSS = post-traumatic stress symptoms. ^a^ Monte Carlo 95% confidence intervals (CI).

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
