# Peer review of "Racial/Ethnic Discrimination and Intimate Partner Violence Perpetration in Latino Men: The Mediating Effects of Mental Health"

_ijerph, 2020, doi:10.3390/ijerph17218148_

Round 1
Reviewer 1 Report
“Effects of Racial/Ethnic Discrimination and Mental Health on Male-Perpetrated Intimate Partner Violence Among U.S.-born and Immigrant Latino Men”
The basic idea of the inquiry is highly significant. The article is very well framed and the research process is presented in scientific and systematic way. I would, however, like to raise some points that may be considered to further improve the manuscript.
- Title: The given title of the research article is a bit lengthy and ambiguous, please make it clear and concise. My suggestions:
“Racial/Ethnic Discrimination and Intimate Partner Violence: The Mediating Effects of Mental Health” or
“The Effects of Racial/Ethnic Discrimination on Male-Perpetrated Intimate Partner Violence: The Mediating Effects of Mental Health”
- Abstract: Overall the abstract is presented in a very good way, I however propose to provide research background in a sentences before the purpose, to complete the full meaning of the abstract, that why the authors are inquiring this issue would be great. For example: Intimate partner violence is a serious public health problem all over the world in respect to racial discrimination and more specifically in the U.S. while the purpose of the article should be start like therefore, the basic purpose of the given study is to test whether racial/ethnic discrimination among….
- Introduction: In the introduction in line 69 the authors presented the heading “Mental Health and Intimate Partner Violence Perpetration” I think this could be presented after the heading “Racial/Ethnic Discrimination and Intimate Partner Violence Perpetration” in line 87 to look more scientific.
- Research Gap: At the end of this section the authors have presented the study proposed hypothesis. Such as: 1) racial/ethnic discrimination will be associated with IPV perpetration for both US-born and immigrant Latinos; and 2) racial/ethnic discrimination will be associated with poorer MH, which in turn will be associated with IPV perpetration. These relationships are expected to be stronger for Latino immigrants than for U.S.-born Latinos.
I recommend that the number of hypothesis should be 3. For example:
Hypothesis 1. Racial/ethnic discrimination will be associated with IPV perpetration for both US-born and immigrant Latinos.
Hypothesis 2. Racial/ethnic discrimination will be associated with poorer MH, which in turn will be associated with IPV perpetration. These relationships are expected to be stronger for Latino immigrants than for U.S.-born Latinos.
Hypothesis 3. MH will mediate the association between racial/ethnic discrimination and IPV perpetration.
- Measures: The study reported about Racial/ethnic discrimination (see line 155) that it was measured by asking participants to report on a 5-point scale later in the last sentence of this paragraph (see line 165 and 166) the author stated that this scale scores ranged from 0 to 4, with higher scores indicating more discrimination. These both statements seem to have some conflicting views please make it clear for the readers.
- Some reference are not as per APA style. Proof reading may also improve the readability of the manuscript. For example, line 155 Experiences of Discrimination scale by Krieger et al., (2005).
- Table 2 is not clear and confusing please make it clear and simple for the readers. It would be great if the authors could provide separate tables for Latino immigrants and U.S-born.
- It is advised to provide a table for indirect effects.
- Discussion: In line 398 to 400 “This was contrary to our expectation that this relationship would be significant for both groups. In both groups, the direct effect of discrimination on IPV perpetration was non-significant.” Please provide here any study that support your study results (link between Discrimination and IPV).
- Last but most important is to provide your own justification in all the proposed hypothesis, as the author has presented only the previous scientific literature but have not presented his own views in the development of hypothesis. Therefore, it is recommended to provide your own ideas and link that to the existing literature would be great.
Good Luck.
Author Response
Title:
- The given title of the research article is a bit lengthy and ambiguous, please make it clear and concise. My suggestions: “Racial/Ethnic Discrimination and Intimate Partner Violence: The Mediating Effects of Mental Health” or “The Effects of Racial/Ethnic Discrimination on Male-Perpetrated Intimate Partner Violence: The Mediating Effects of Mental Health”
As suggested by Reviewer 1, we have changed the title to “Racial/Ethnic Discrimination and Intimate Partner Violence Perpetration in Latino Men: The Mediating Effects of Mental Health.”
Abstract:
- Overall the abstract is presented in a very good way, I however propose to provide research background in a sentences before the purpose, to complete the full meaning of the abstract, that why the authors are inquiring this issue would be great. For example: Intimate partner violence is a serious public health problem all over the world in respect to racial discrimination and more specifically in the U.S. while the purpose of the article should be start like therefore, the basic purpose of the given study is to test whether racial/ethnic discrimination among….
Please note that we have added a sentence in the abstract that broadly speaks to the background. However, as a result, we have surpassed the maximum word count for the abstract by 34 words.
Introduction:
- In the introduction in line 69 the authors presented the heading “Mental Health and Intimate Partner Violence Perpetration” I think this could be presented after the heading “Racial/Ethnic Discrimination and Intimate Partner Violence Perpetration” in line 87 to look more scientific.
We agree with Reviewer 1 and have now moved the “Mental Health and Intimate Partner Violence Perpetration” section (now line 82) after the “Racial/Ethnic Discrimination and Intimate Partner Violence Perpetration” section (now line 70).
- At the end of this section the authors have presented the study proposed hypothesis. Such as: 1) racial/ethnic discrimination will be associated with IPV perpetration for both US-born and immigrant Latinos; and 2) racial/ethnic discrimination will be associated with poorer MH, which in turn will be associated with IPV perpetration. These relationships are expected to be stronger for Latino immigrants than for U.S.-born Latinos. I recommend that the number of hypothesis should be 3. For example:
Hypothesis 1. Racial/ethnic discrimination will be associated with IPV perpetration for both US-born and immigrant Latinos.
Hypothesis 2. Racial/ethnic discrimination will be associated with poorer MH, which in turn will be associated with IPV perpetration. These relationships are expected to be stronger for Latino immigrants than for U.S.-born Latinos.
Hypothesis 3. MH will mediate the association between racial/ethnic discrimination and IPV perpetration.
The hypotheses have been edited to reflect the three hypotheses recommended by Reviewer 1 (lines 150-157).
Methods:
- The study reported about Racial/ethnic discrimination (see line 155) that it was measured by asking participants to report on a 5-point scale later in the last sentence of this paragraph (see line 165 and 166) the author stated that this scale scores ranged from 0 to 4, with higher scores indicating more discrimination. These both statements seem to have some conflicting views please make it clear for the readers.
We believe there may have been a misunderstanding. We have clarified the scoring to reduce confusion on line 200-201 to read that the response options ranged: “from “never” (0) to “very often” (4).
References:
- Some reference are not as per APA style. Proof reading may also improve the readability of the manuscript. For example, line 155 Experiences of Discrimination scale by Krieger et al., (2005).
We have reformatted the references for consistency with IJERPH style
Tables:
- Table 2 is not clear and confusing please make it clear and simple for the readers. It would be great if the authors could provide separate tables for Latino immigrants and U.S-born.
Table 2 has been reformatted to include the names of the variables as well as their abbreviations. We believe this makes the table clearer to readers. We believe it is best to present both groups together as it is easier for readers to directly compare the two groups when they are in the same table. We have added a line or border to separate the two groups and have also modified the title to make it clearer that there is data on both groups.
- It is advised to provide a table for indirect effects.
As suggested by Reviewer 1, a table (labelled Table 3) for the indirect effects has been added at line 366.
Discussion:
- In line 398 to 400 “This was contrary to our expectation that this relationship would be significant for both groups. In both groups, the direct effect of discrimination on IPV perpetration was non-significant.” Please provide here any study that support your study results (link between Discrimination and IPV).
We have cited a study to support our study results on lines 422-429. As noted, the research examining the link between discrimination and IPV is limited so we compare our findings among Latinos with the one study we were able to find that also examined these factors among Latinos.
Hypotheses:
- Last but most important is to provide your own justification in all the proposed hypothesis, as the author has presented only the previous scientific literature but have not presented his own views in the development of hypothesis. Therefore, it is recommended to provide your own ideas and link that to the existing literature would be great.
We have added, in our own words, how we developed our hypotheses on line 129-137.
Reviewer 2 Report
It was a pleasure having the opportunity to review this manuscript After carefully reviewing this manuscript, I find that there are some areas which need to be addressed before the paper is ready for publication. The formulated suggestions are meant to support further development of the manuscript.
General thoughts. The work contains defects.
1. The severe weakness of the text. The goal is NOT specified explicitly enough. Authors should explain why the document was prepared, who should profit from the new knowledge that is acquired in their investigations. Unfortunately, Authors fail to specify, WHICH TYPE OF BENEFITS, and for WHAT TYPE OF STAKEHOLDERS, the description of their inquiry results may be of interest or support the managerial and policy decisions.
2. The weakness of the text. There is a lack of data description. Authors fail to describe the respondents, demographic and socioeconomic characteristics is missing,in paragraphs 145 to 151. Paragraphs 252 to 262 should be in the sample section and not in the results.
Generally speaking, authors should elaborate description of the data collection process and data set composition.
3. The weakness of the text. Authors do not provide enough literature references concerning similar data analysis. This fact, combined with vague, imprecise statements about the analytical techniques used in the study of the collected data, makes it not easy to assess whether the methods used for data analysis are appropriate
4. Authors should try and distinguish, which statements are Authors' opinion, what is the literature knowledge and what comes as the analysis outcome. The used literature references are (almost exclusively) referred to in such a way, that it is not clear why the publication is cited. Usually, there are no details, whether individual, mentioned authors support Authors theses and findings. Authors should precisely specify which references support their position and why, which are in opposition to their conclusions and why. There is no generalisation effort in the literature review. Besides the fact that the references are in most cases too old (paragraphs 55 to 66,116-125 of more than the last 5 years)
Authors should reformulate the text should of literature review.
The style of the text, instead of being purely reporting, descriptive, needs to be analytical, with generalising indications.
5. In the results the tables do not follow the format of the journal and lack important results such as explaining why the parametric have been used (M of Box, etc...) in addition it is necessary to include the eta square, for further clarification.
6. In the structural one it is necessary to include more parameters NFI, CMI, PNFI, etc and the comparative table of both models.
7. The severe weakness of the text. In part entitled CONCLUSIONS, the conclusions are NOT included. It is merely a description of the submission text. Authors repeated analogous description from the introductory parts of the article. The formulation of conclusions should contain elements which are anchored in research findings. They should be based on analytical statements describing the MERIT TOPIC analysis results.
Author Response
Reviewer 2
- The severe weakness of the text. The goal is NOT specified explicitly enough. Authors should explain why the document was prepared, who should profit from the new knowledge that is acquired in their investigations. Unfortunately, Authors fail to specify, WHICH TYPE OF BENEFITS, and for WHAT TYPE OF STAKEHOLDERS, the description of their inquiry results may be of interest or support the managerial and policy decisions.
Please note that we have added several sentences starting at line 58 and 62 that further clarify why this research was conducted. We also added a sentence on line 67-69 which clearly denotes to whom this research would be relevant and why. We previously had directly noted mental health professionals and providers of IPV services in the implications section. However, we have now also added primary care professionals to the implications section on line 505-507. Also, we added a final paragraph in the Introduction (line 158-164) that addresses directly Reviewer 2’s comments by including the potential results, benefits and stakeholders who might be interested.
- The weakness of the text. There is a lack of data description. Authors fail to describe the respondents, demographic and socioeconomic characteristics is missing in paragraphs 145 to 151. Paragraphs 252 to 262 should be in the sample section and not in the results.
Generally speaking, authors should elaborate description of the data collection process and data set composition.
We have moved the demographics information to the participants subsection of Methods section (now line 174 to 187). We have clarified that the current study is a secondary analysis by adding this information on line 167 and refer the reader to published sources for information on sampling and data collection (line 171-172). In the text for this section, we now include demographic and sociodemographic characteristics (unweighted) for the sample.
- The weakness of the text. Authors do not provide enough literature references concerning similar data analysis. This fact, combined with vague, imprecise statements about the analytical techniques used in the study of the collected data, makes it not easy to assess whether the methods used for data analysis are appropriate
To address Reviewer 2’s concerns regarding our data analytic approach, we now provide a rationale for selection of the structural equation modelling approach (a commonly used statistical technique for mediational model testing) and the advantages of this approach.
- Authors should try and distinguish, which statements are Authors' opinion, what is the literature knowledge and what comes as the analysis outcome. The used literature references are (almost exclusively) referred to in such a way, that it is not clear why the publication is cited. Usually, there are no details, whether individual, mentioned authors support Authors theses and findings. Authors should precisely specify which references support their position and why, which are in opposition to their conclusions and why. There is no generalisation effort in the literature review. Besides the fact that the references are in most cases too old (paragraphs 55 to 66,116-125 of more than the last 5 years). Authors should reformulate the text should of literature review.
The style of the text, instead of being purely reporting, descriptive, needs to be analytical, with generalising indications.
It was not clear to us if these comments referred to the Discussion section, the Introduction section, or both. We assumed that all of these comments were referring to the Introduction. If this is not the case, please clarify further. For paragraphs 55 to 66, 116-125 (which are now lines 53-57 and 117-127), we searched the literature and provided the most recent references. If the reviewer has more recent citations that they believe we missed, please include those so that we may update our references as we were still unable to locate other citations in the past 5 years (besides Okuda et al., 2015). Consistent with the APA Publication Manual (6th ed.), we do not cite our own ideas, conclusions and common knowledge; any text that includes citations describes the ideas of the authors cited.
- In the results the tables do not follow the format of the journal and lack important results such as explaining why the parametric have been used (M of Box, etc...) in addition it is necessary to include the eta square, for further clarification.
Regarding Table 1, it appears that Reviewer 2 is mentioning statistics that are typically reported with ANOVA’s (i.e. M of Box and eta square). Please note that Table 1 is a t-test. We have added a footnote to Table 1 that specifies that the p-value if for a t-test for group differences between immigrants and U.S.-born Latinos. We also added another note to Table 1 to clarify that the values take into account the sampling design, which is why we used population parameters (i.e. SE) rather than sample statistics (i.e. SD). We have reformatted Table 2 to have clearer headings on line 310.
- In the structural one it is necessary to include more parameters NFI, CMI, PNFI, etc and the comparative table of both models.
Unfortunately, the lavaan package in R does not provide the fit indices recommended by the Reviewer. However, please note that other fit indices have already been reported in lines 331-332. We believe that those fit indices reported are sufficient as they are recommended and cited repeatedly by prominent researchers (see Hooper et al., 2008). We also added the fit indices to the figures per the Reviewer’s recommendation.
Although a table was not added for the fit statistics of the unconstrained and constrained models, the fit indices for the fully constrained model were added to the text on line 332-333. We opted for including the information in the text, as opposed to a table format, since the information is relatively brief and easy to include in text and because the fit statistics for the unconstrained model were already in the text.
Hooper, D, Coughlan, J and Mullen, M (2008) Structural Equation Modelling: Guidelines for Determining Model Fit. Electronic
Journal of Business Research Methods, 6(1), 53-60
- The severe weakness of the text. In part entitled CONCLUSIONS, the conclusions are NOT included. It is merely a description of the submission text. Authors repeated analogous description from the introductory parts of the article. The formulation of conclusions should contain elements which are anchored in research findings. They should be based on analytical statements describing the MERIT TOPIC analysis results.
We agree with Reviewer 2. We have edited the Conclusions sections (lines 552-566) to reflect general conclusions that are anchored in the findings rather than repeating a summary of the findings.
Round 2
Reviewer 1 Report
Dear authors, overall you people have done an excellent work in the updated version of the manuscript. However, I have some minor suggestions about study title and table 2:
- Title: The updated version of the study title exhibit only about Latino Men while the study had also inquired about US born inhabitants which is missing in the title. My suggestion in this regard is, either don't mention about Latino or US born as it is stated in the study abstract, introduction and later in methodology as well. So, there is no need to mention it in title, if the authors believe that it should be mention in the title then better to include both (Latino Men and US born) not only Latino.
- Table 2 has been reformatted which looks great and is more clear than previous version. The reviewer is agreed with the authors response “that it is best to present both groups together as it is easier for readers to directly compare the two groups when they are in the same table”. However, the line drawn by the authors in the table looks not very good and it is advised to remove the line from the table.
Best of luck…
Author Response
- Title: The updated version of the study title exhibit only about Latino Men while the study had also inquired about US born inhabitants which is missing in the title. My suggestion in this regard is, either don't mention about Latino or US born as it is stated in the study abstract, introduction and later in methodology as well. So, there is no need to mention it in title, if the authors believe that it should be mention in the title then better to include both (Latino Men and US born) not only Latino.
Because the entire sample consists of Latino men, we feel it is important to mention them in the title. Therefore, we have decided to retain “Latino men” in the title. Please note that we use the term “Latino” to reflect both individuals who were born in Latin America and their descendants. Therefore, this includes any person born in the U.S. whose ancestors (e.g., parents and grandparents) are from Latin America. This is also the definition used by the U.S. Census and that used by the NESARC study, from which we obtained our sample.
- Table 2 has been reformatted which looks great and is more clear than previous version. The reviewer is agreed with the authors response “that it is best to present both groups together as it is easier for readers to directly compare the two groups when they are in the same table”. However, the line drawn by the authors in the table looks not very good and it is advised to remove the line from the table.
We agree with the reviewer’s comment about the appearance of Table 2 and have removed the line.
Reviewer 2 Report
I appreciate the improvements made to the document and believe that it has enhanced its quality. However, I believe that they do not follow the rules of the magazine in terms of format so I recommend the revision of them as minor improvements.
Author Response
- I appreciate the improvements made to the document and believe that it has enhanced its quality. However, I believe that they do not follow the rules of the magazine in terms of format so I recommend the revision of them as minor improvements.
We based our formatting on the “Instructions to Authors” and updated the references to reflect that of the journal’s preferred style. However, without further clarification, we are not aware of what other formatting rules the reviewer believes we are not following. We will work with the editors to address any formatting issues that they come across.